# Study on surface deformation induced by shield excavation due to coarse particle content in strongly weathered rock layers

**Puzhen An[1], Baoxin Jia[1,2]\*, Qinglei Yuan[1], Lei Chen[1,2]**

**1** School of Civil Engineering, Liaoning Technical University, Fuxin, Liaoning, China, **2** Key laboratory of Disaster Management and Ecological Restoration in Resource-depleted Mining areas of Liaoning Province, Fuxin, Liaoning, China

\* cd919419842@163.com

## Abstract

Currently, there is limited research focusing on the surface deformation caused by shield excavation in strongly weathered rock formations with significant coarse particle content. This study, based on the shield tunnel project of the Pearl River Delta Intercity Pazhou Branch Line, investigates the influence of coarse particle content in strongly weathered rock layers on their mechanical properties, as well as its effect on surface subsidence during tunnel shield excavation. By comparing Peck's formula with random medium theory, the study expands the application of random medium theory in engineering, converting many of Peck's empirical parameters into corresponding parameters within the random medium framework. Additionally, Peck's formula is modified to account for the coarse particle content in strongly weathered rock layers. The results show that as the coarse particle content increases, the force transmission mechanism in the strongly weathered rock shifts from compressive deformation typical of fine-grained soils to point-contact force transmission between coarse particles. Furthermore, as the coarse particle content rises, the strata loss rate tends to decrease gradually. The final settlement curve, calculated using the method that considers changes in coarse particle content, is closer to the measured values. This study's calculation method more accurately reflects the surface deformation behavior caused by shield tunnel construction in strongly weathered rock layers, particularly when coarse particle content is taken into account. It also provides a better understanding of the combined impact of various construction parameters as the coarse particle content changes during the excavation process.

## 1 Introduction

With the rapid development of rail transit in China, the shield construction technique (SCT) has been widely used in the construction of urban subway tunnels because of its advantages such as little impact on the surrounding environment, wide range of ground adaptation, and fast construction speed [1]. For the surface deformation caused by shield tunnel construction, scholars have conducted multi-dimensional studies, and one is to predict the surface settlement by modifying the traditional Peck equation [2]. Song et al. [3] fitted Peck's formula

using the least squares optimization method based on the on-site monitoring values of the underground comprehensive pipeline corridor constructed by the shield method. Yan et al. [4] used Peck's formula for surface settlement prediction, due to the large error between Peck's predicted data and measured data, the maximum settlement correction factor and settlement correction factor were introduced to correct peck's formula. Wang et al. [5] predicted the surface settlement of subway double line tunnel in shallow buried loess area. Yang et al. [6] used Peck's correction formula to predict the surface settlement of shield tunnel in water-rich sand and pebble stratum. Zhang et al. [7] established a surface settlement model to solve the problem of increasing surface settlement with time. Considering the effects of time factor and depth on surface settlement, the reliability of the proposed modified Peck formula for calculating construction-induced surface settlement in shallow buried tunnels is verified on the basis of comparison between FLAC 3D calculation software and field monitoring. Yin et al. [8] provided a method for obtaining Peck equation parameter values under different engineering conditions. Zhang et al. [9] improved Peck equation based on data from several field cases, and the calculated results were in good agreement with the field measurement results. Huang et al. [10] used different prediction methods to fit and analyze the transverse and longitudinal ground settlement for the soft soil shield tunnel. Liu et al. [11] modified Peck equation by linear regression method and fitted the curve of tunnel cross section surface settlement with high consistency with the actual data. However, due to the large regional differences and limitations in the application of this equation, the essence is still the lack of theoretical basis for experience summary.

Another method for predicting land subsidence is the stochastic medium theory, which has a clear theoretical derivation process. Wang et al. [12] investigated the prediction of surface settlement caused by shield construction in sandy pebble strata based on random medium theory. Wei et al. [13] derived the formula for lateral surface settlement caused by QRS tunnel boring based on random medium theory. The palm face convergence mode coefficient $\alpha$ is introduced to make the derived equation applicable to various palm face convergence modes. Shao et al. [14] derived the non-uniform convergence prediction model for different section forms of tunnels based on the random medium theory method. Zheng et al. [15] established a calculation model of soil deformation caused by tunneling-induced surface loss according to the characteristics of DOT shield boring and crushing. Based on the theory of random medium, the theoretical solution of soil deformation considering the crushing by DOT shield machine was obtained by polar coordinate transformation and multinomial domain integration method. Liu et al. [16] established a surface settlement prediction model considering the tunnel's non-uniform convergence and deformation. Yang et al. [17] used the random medium theory to model the ground movement caused by the tunnel and formulated the ground settlement equation, so as to predict the ground deformation more reliably. Wang [18] compared and analyzed the Peck settling trough equation, which is commonly used to predict surface deformation, with random medium theory, and then converted Peck empirical parameter into the random medium theory parameter, which solved the problem of difficult-to-obtain section convergence area $\Delta A$ and formation main influence angle $\beta$ in traditional random medium theory, and expanded the application range of random medium theory.

In the southeast of China, especially in Guangdong and Fujian provinces, strongly weathered rock strata are widely distributed. Due to the large difference in the degree of softness and hardness of adjacent soil caused by strong weathered rock layers, the tunneling process of the shield machine is extremely difficult during shield construction, and it is easy to cause damage to the tool of the shield machine, which brings great inconvenience to the construction. Domestic and foreign scholars have conducted a large number of studies on the ground settlement caused by shield construction in different regions with different geological

conditions [19–22]. Sun et al. [23] explored the influence of different vertical pressures and coarse particles with different particle sizes, shapes, and locations on the shear strength and migration characteristics of the ore-rock particle system. Hu et al. [24] conducted large-scale direct shear tests on granite residual soil with different coarse particle contents. In addition, Zhang et al. [25] based on artificial neural network and other algorithms, established a prediction model for surface settlement caused by soil pressure balance shield construction with high accuracy; Jia et al. [26] proposed a prediction method for surface deformation caused by curvilinear shield tunneling in upper-soft and lower-hard strata; Zhang et al. [27] finite element analysis was conducted to study the ground and tunnel response to partial tunnel leakage coupled with anisotropic soil permeability.

At present, there is not enough research on the effect of coarse particle content in strongly weathered rock layers on the surface deformation caused by shield excavation. In order to better elucidate the role of coarse particle content on surface deformation during shield construction, the author combines Peck's formula with the commonly used surface deformation analysis methods in random medium theory, and based on the shield tunnel project example of the Pearl River Delta Intercity Pazhou Branch Line, the author thoroughly discusses the surface deformation law triggered by shield excavation by the coarse particle content of strongly weathered rock layers. By comprehensively considering the influence of the change of coarse particle content on construction parameters, it aims to provide a more perfect theoretical and practical basis for the design of shield tunnel construction and surface deformation control in strongly weathered rock formations.

## 2 Effect of coarse particle content on mechanical properties of strongly weathered rock

### 2.1 Effect of coarse particle content on compaction characteristics

Weathered granite is a kind of coarse-grained material with complex properties that are related to the composition of soil particles and the structure of soil. Thus, strongly weathered granite can be considered as a coarse-grained soil material. Coarse particles in strongly weathered rocks are those whose particle size exceeds 10mm, and the content of coarse particles determines the contact state and strength characteristics of particles in strongly weathered rocks. When the proportion of coarse particles is relatively low, the strongly weathered rock shows a suspended state, and the coarse particles are wrapped by finer particles, which results in the particle skeleton cannot directly contact to and transfer the load. In this case, the compaction work is mainly dissipated through the dense deformation of fine granular soil. At this time, the strength of strongly weathered rock is mainly determined by the density degree of particles. Moreover, when there are a lot of coarse particles in the soil, the particles are in interstice contact. The coarse particles form inner skeleton contact, and the interstice is filled by fine particles, partly filled by crushed coarse particles. This kind of weathered rock filler mainly relies on the crushing, wear, and displacement of coarse particles to consume the compaction work. Its compaction law is different from the traditional soil, and also different from the hard stone slag filler. Fig 1 shows the compaction characteristic curves of strongly weathered rocks with different coarse particle content.

As can be seen from Fig 1, the optimal water content is 13.95% and 9.30% when the coarse particle content in the strongly weathered rock is 20% and 80%, respectively, showing that with the increase of coarse particle content, the maximum water content decreases gradually. When the coarse particle content of strongly weathered rock reaches 30%, the dry weight reaches the maximum value of 2.06g/cm³; and when the coarse particle proportions are 40% and 80%, the dry weight is 2.05g/cm³. Furthermore, when the coarse particle content in the

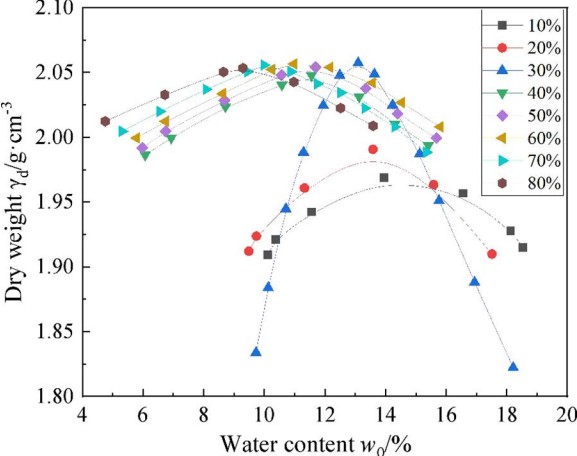

**Fig 1. Compaction characteristic curve of strongly weathered rock with different coarse particle contents.**

soil is 20%, the dry weight reaches the minimum value of 1.99g/cm³, which is the smallest in all the coarse particle ratios of strongly weathered rocks. The reason is that the ratio of coarse particles affects the force transfer mode of particle skeleton in strongly weathered rock. More specifically, when the coarse particle content is 20%, the coarse particles are suspended in the fine particles of strongly weathered rock, and the dry weight mainly depends on the compaction characteristics of the fine particles. Under the action of the compaction load, the fine particles are compacted, and the compaction work is mainly dissipated by the friction between the particles, resulting in the maximum dry density being only 1.99g/cm³ at this time. This kind of strongly weathered rock has a relatively large cohesive force, relatively small internal friction angle, and poor permeability. When the coarse particle content is 80%, the coarse particles are in direct contact with each other, and the coarse particles are crushed under the action of compaction load, which means that the compaction work is mainly converted into the broken surface energy of the coarse particles and the particle slip dissipation energy. At this time, the maximum dry density is only 2.05g/cm³.

When the coarse particle content is 30%, the coarse particles present a point contact state, and the fine particles fill the pores. Under the action of external loads, both the coarse particles will be broken and the fine particles will slip, and the compaction work will be transformed into both the surface energy of the broken coarse particles and the dissipated energy of the slip between the fine particles. Moreover, when the coarse particle content in the soil is 30%, the γd value is the largest, because the small particles and fine materials are fully filled in the void of the coarse particles, showing an ideal dense skeleton structure. The dry weight of strongly weathered rock at this time depends on the characteristics of the coarse particles and fine materials, and the porosity in the mixture is the smallest. At this time, the values of strongly weathered rocks are higher, and the strength and water stability are better. However, the compaction curve of this kind of soil is steeper and the variation range of water content is smaller, so it is difficult to control the optimal water content in construction.

When the coarse particle content of strongly weathered rock exceeds 30%, the skeleton void structure appears, and the maximum dry bulk weight does not increase with the increase of the coarse particle content. On the contrary, there are more voids in the coarse particles, and the content of fine materials is not enough to completely fill the voids, as the size of the dry weight mainly depends on the characteristics of the coarse particles. Although the bulk

density of the coarse particles is larger, the dry weight is still low due to more voids. This kind of weathered rock packing has a large internal friction angle after compaction, but the cohesion is very low, and it is easy to seepage water.

## 2.2 Effect of coarse particle content on consolidation characteristics

In the consolidation test, floating ring compressors were used to test the compression characteristics of strongly weathered rocks with different coarse particle content. The test load classes were 0.05MPa, 0.1MPa, 0.2MPa, 0.4MPa, 0.8MPa, 1.6Mpa, and 3.2MPa, respectively. The deformation of the sample was measured by a dial indicator, and after the deformation was stable, unloading was carried out step by step. According to the experimental data, the pore ratio and load relationship (e ~ P) curves were drawn to determine the compression characteristics of strongly weathered rocks. The curves of strongly weathered rock samples with different coarse particle contents under various loads are shown in Fig 2. Besides, the compression characteristic indexes of strongly weathered rocks with different coarse particle content can be obtained from the compression characteristic curves of strongly weathered rocks, and the calculation results are shown in Table 1.

As shown in Table 1, the dry density of the sample ranges from 2.04 to 2.05 g·cm$^{-3}$, the average compression coefficient ranges from $1.97 \times 10^{-5}$ to $2.48 \times 10^{-5}$, and the compression modulus ranges from $1.48 \times 10^{5}$ to $1.98 \times 10^{5}$. Thus, with the increase of coarse particle content in strongly weathered rock, the dry density and compression coefficient tend to decrease, and the compression modulus tends to increase. When the coarse particle content is 20%, the dry density and compression coefficient of the sample reach the maximum value, and the compression modulus reaches the minimum value, indicating that the lower the coarse particle content, the worse the deformation resistance of strongly weathered rock. This is mainly because when the coarse particle content is low, a complete particle skeleton cannot be formed inside the sample, which refers to the compression characteristics of fine particle soil.

## 2.3 Effect of coarse particle content on strength

The TW-2000 triaxial tester was selected for experiments in this study. The diameter and height of samples were 50mm and 100mm, respectively. The surface vibration method was used to load the sample, and the confining pressure was selected as 100 kPa, 200 kPa, 300 kPa, and 400 kPa, respectively. The test was carried out in accordance with the relevant requirements of the "Standard for Geotechnical Test Methods (GB/T50123-2019)". Table 2 shows the triaxial test results of strongly weathered rocks with different coarse particle contents (10%, 20%, 30%, 40%, 50%, 60%, 70%, and 80%). Fig 3 shows the molar envelope of shear strength parameters of strongly weathered rocks with different coarse particle contents.

The shear strength parameters of strongly weathered rock can be determined according to the slope and intercept of the envelope of shear strength. The intercept corresponds to the cohesion of strongly weathered rock, and the slope corresponds to the Angle of internal friction. The calculation results of cohesive force and internal friction angle of strongly weathered rocks with different coarse particle contents are shown in Table 3. As shown in Table 3, with the increase of coarse particle content, the cohesion gradually decreases from 37.28kPa to 22.65kPa, and the internal friction angle gradually increases from 27.65° to 36.36°. Besides, when the coarse particle content increases from 20% to 80%, the force transfer mode of strongly weathered rock is transformed from the compression deformation of fine soil to the point contact force transfer between coarse particles. The mechanical intercalation and friction between coarse particles are related to the irregular geometric shape characteristics of coarse particles, the content of needle particles, and the grading characteristics. Therefore,

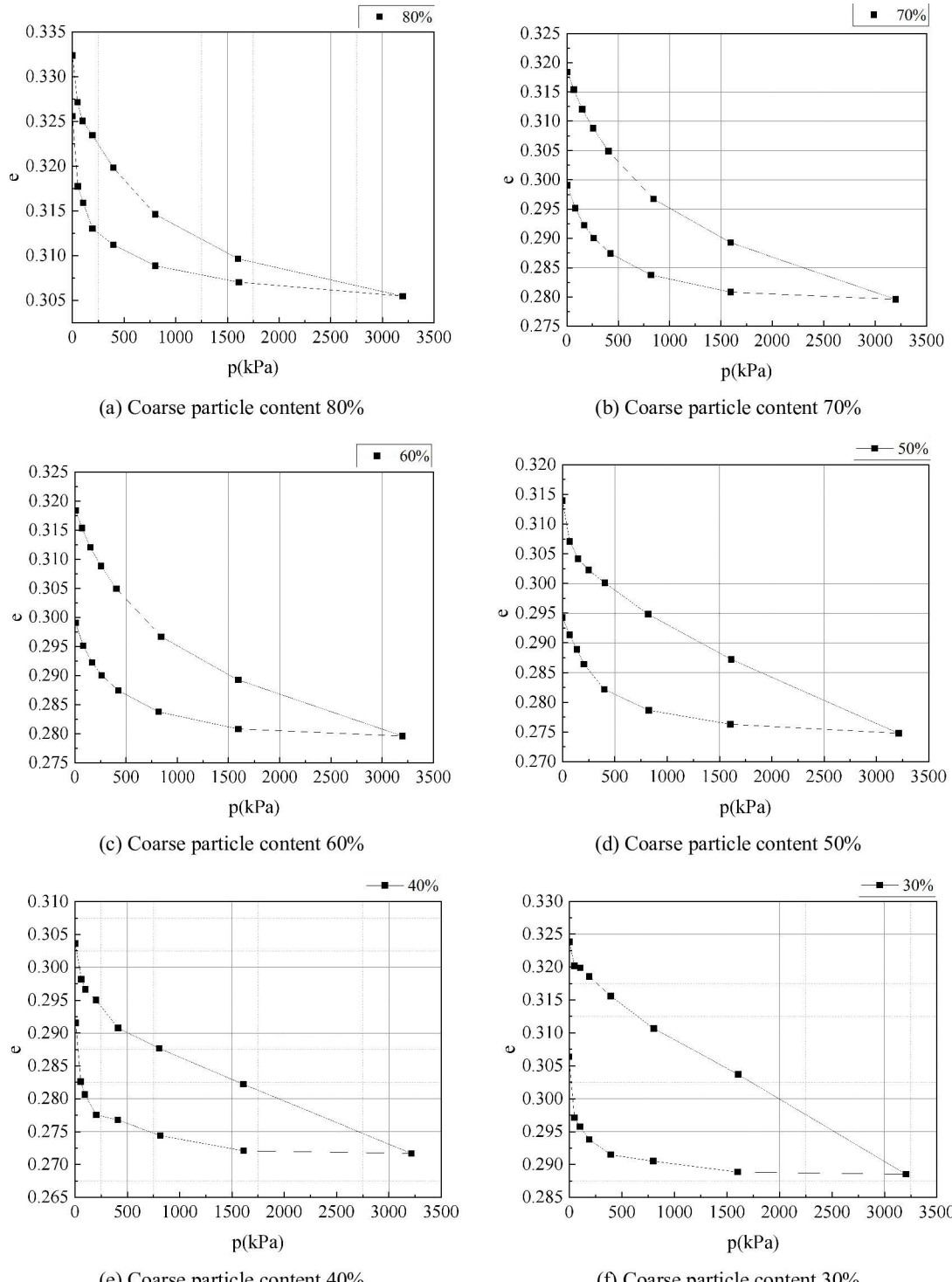

Fig 2. Compression characteristic curve of strongly weathered rock with different coarse particle content.

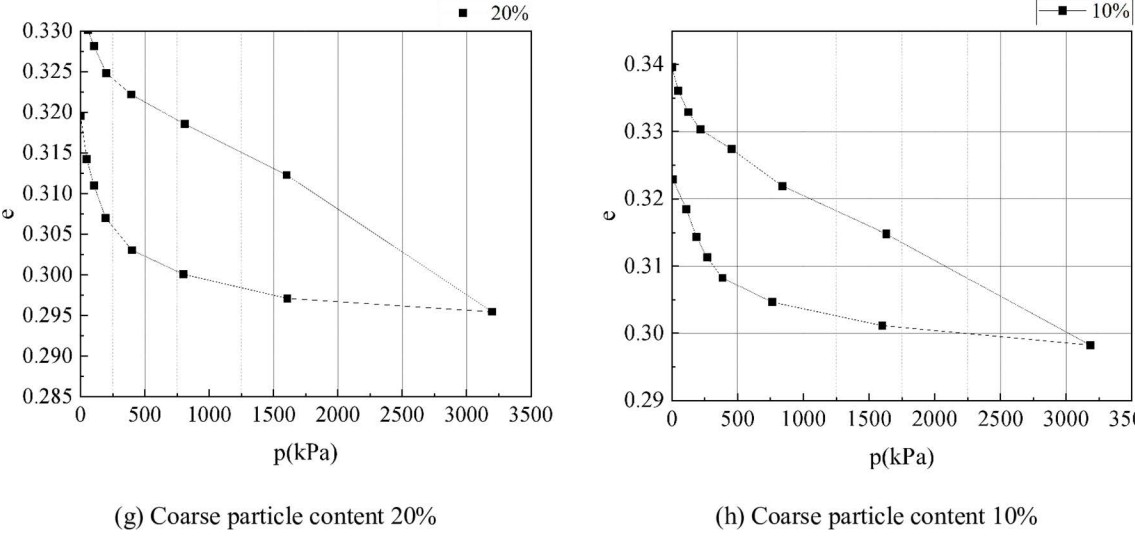

(g) Coarse particle content 20%    (h) Coarse particle content 10%

**Fig 2.** Continued.

**Table 1. Compressive property indicators of strongly weathered rock.**

| Coarse particle content | Water content of sample/ % | Dry density of sample $\rho_d$/ (g·cm⁻³) | Compression coefficient $a_v$/ (kPa⁻¹) | Compression modulus $E_s$/ kPa |
|---|---|---|---|---|
| 80% | 6.2 | 2.04 | 1.62E-05 | 61728.40 |
| 70% | 6.3 | 2.08 | 1.68E-05 | 59523.81 |
| 60% | 6.5 | 2.06 | 1.96E-05 | 51020.41 |
| 50% | 6.4 | 2.05 | 2.36E-05 | 42372.88 |
| 40% | 6.4 | 2.05 | 2.54E-05 | 39370.08 |
| 30% | 6.5 | 2.07 | 2.87E-05 | 34843.21 |
| 20% | 6.5 | 2.08 | 3.47E-05 | 28818.44 |
| 10% | 6.5 | 2.11 | 3.51E-05 | 27251.32 |

**Table 2. Major Principal Stress $\sigma_1$ Results of Triaxial Test for Strongly Weathered Rocks with Different Coarse Particle Content.**

| Coarse particle content | Confining pressure/ kPa | | | |
|---|---|---|---|---|
| | 100 | 200 | 300 | 400 |
| 80% | 418.39 | 727.00 | 1035.60 | 1344.20 |
| 70% | 433.69 | 756.46 | 1079.23 | 1402.00 |
| 60% | 441.46 | 772.17 | 1102.87 | 1433.58 |
| 50% | 461.97 | 811.21 | 1160.45 | 1509.69 |
| 40% | 494.65 | 874.50 | 1254.35 | 1634.20 |
| 30% | 522.74 | 929.36 | 1335.99 | 1742.61 |
| 20% | 537.39 | 961.04 | 1384.69 | 1808.34 |
| 10% | 552.05 | 992.68 | 1433.31 | 1873.93 |

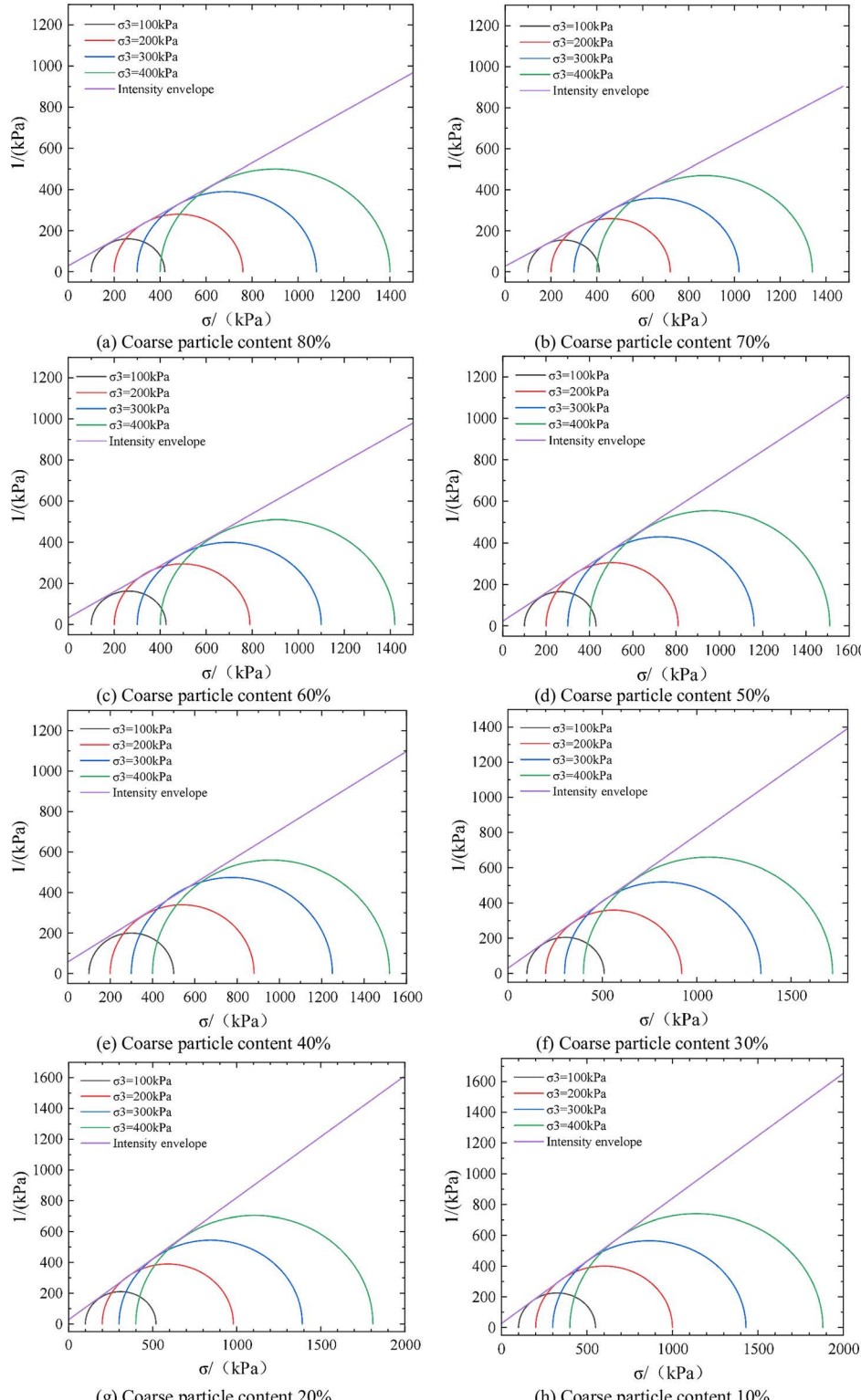

**Fig 3. Experimental results of shear strength of strongly weathered rocks with different coarse particle contents.**

**Table 3. Calculation results of shear strength parameters for strongly weathered rocks with different coarse particle contents.**

| Coarse particle content | Shear strength envelope equation | Cohesion $c$/ kPa | Internal friction angle $\varphi$/ (°) |
|---|---|---|---|
| 80% | $y = 0.5948x + 31.25$ | 31.25 | 30.76 |
| 70% | $y = 0.6211x + 30.87$ | 30.87 | 31.86 |
| 60% | $y = 0.6354x + 30.45$ | 30.45 | 32.45 |
| 50% | $y = 0.6680x + 30.16$ | 30.16 | 33.76 |
| 40% | $y = 0.7192x + 29.45$ | 29.45 | 35.74 |
| 30% | $y = 0.7616x + 28.79$ | 28.79 | 37.31 |
| 20% | $y = 0.7875x + 27.63$ | 27.63 | 38.24 |
| 10% | $y = 0.8127x + 26.54$ | 26.54 | 39.12 |

when the shield tunnel is excavated, it is necessary to pay attention to the coarse particle content in strongly weathered rock to adjust the shield parameters in time according to the difference in formation properties.

## 2.4 Influence on resilience modulus

The resilience modulus was determined by a triaxial compression test under recirculating loading. The sample size was 50mm × 100mm, and the amplitude of cyclic loading-unloading was selected as $1/3P_{max}$ ($P_{max}$ was the peak load). The reason for selecting this stress interval was that the sample was under elastic deformation in this stress interval, and no bulk strain would occur. However, when the stress was between $1/3P_{max}$ and $2/3P_{max}$, the volume deformation of the sample changed from volume contraction to volume expansion, and hence the confining pressure during the test was selected as 100 kPa, 200 kPa, 300 kPa, and 400 kPa, respectively.

The loading rate has a significant impact on the resilience modulus of strongly weathered rock. In this study, a constant loading rate of 1.0mm/min is adopted to carry out loading-unloading tests on the samples of strongly weathered rock. Since the dry weight of strongly weathered rock with a coarse particle content of 30% is significantly affected by the change in water content, the water content was not variated during loading. When the axial strain is loaded to 0.5%, the unloading begins until the principal stress difference is zero, and then the loading and unloading restart until the sample is damaged. The principal stress difference and axial strain relationship curves of the weathered rock sample with strong coarse particle content are shown in Fig 4. As can be seen from Fig 4, with the increase of axial stress, the axial stress difference and resilience modulus of strongly weathered rock samples gradually increase. When the confining pressure increases to 100 kPa, the uniaxial compressive strength and resilience modulus increase to 735.58 kPa and 220.12 MPa, respectively. When the confining pressure increases to 400 kPa, the uniaxial compressive strength and resilience modulus increase to 1415 kPa and 242.47 MPa, respectively. Thus, increasing confining pressure can effectively improve the resilience and deformation resistance of strongly weathered rock.

## 3 Effect of coarse particle content on formation loss in tunnel shield excavation

Peck proposed a formula for estimating transverse land settlement under undrained conditions as below [28]:

$$S(x) = S_{max} \exp\left[-\frac{x^2}{2i^2}\right]$$

(1)

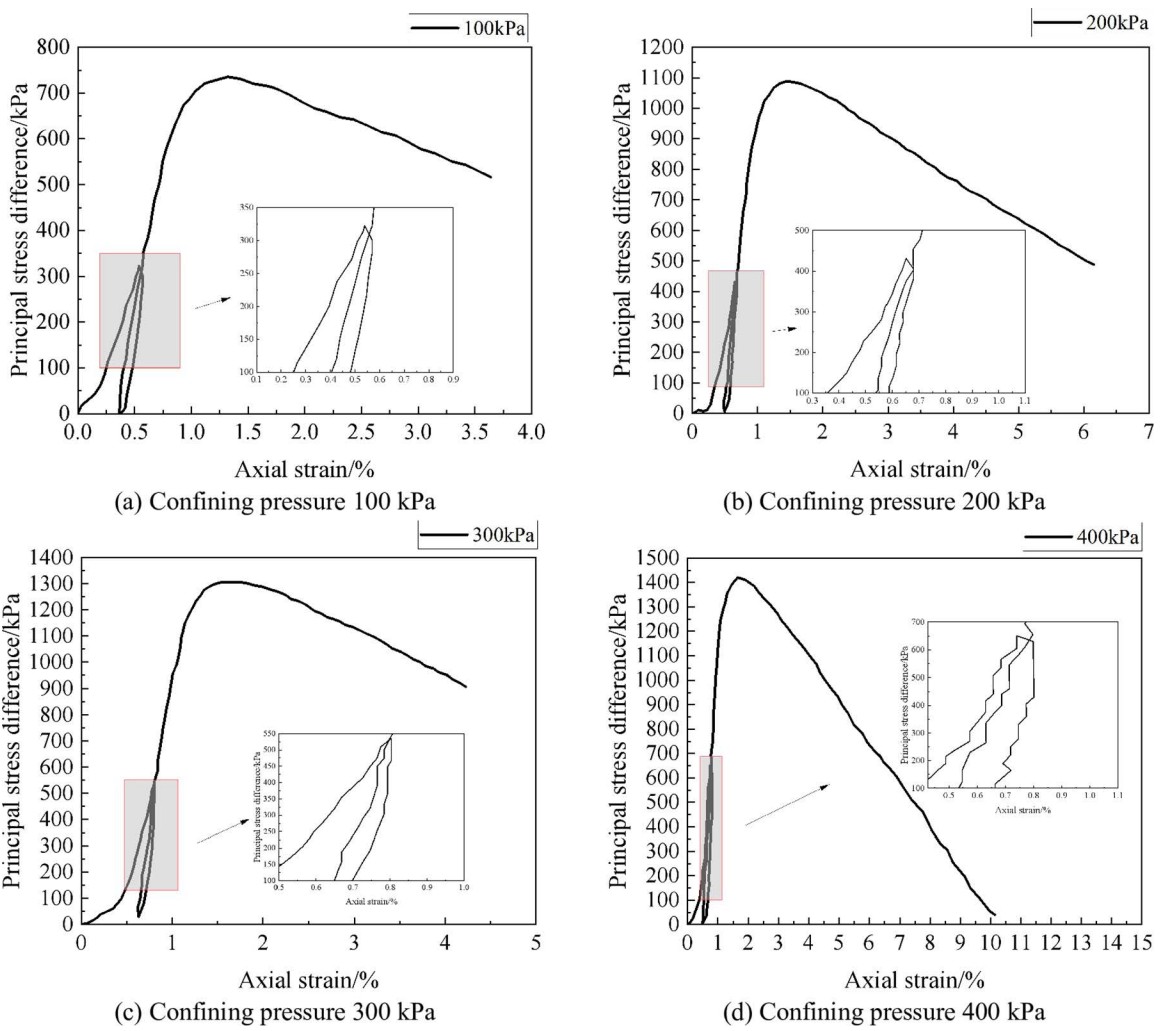

**Fig 4. Relationship Curve between Principal Stress Difference and Axial Strain of Strongly Weathered Rock.**

$$S_{\max} = \frac{V_s}{i\sqrt{2\pi}} \tag{2}$$

$$V_1 = \frac{V_s}{\pi R^2} \tag{3}$$

where, $S(x)$ is the vertical settlement of the ground at the calculation point x, and x is the horizontal distance from the calculation point to the center of the tunnel; $S_{\max}$ is the maximum vertical settlement of the ground above the midpoint of the tunnel; $i$ is the width of the settlement tank, and $i = kh$; $k$ is the coefficient of the ground settlement trough, and $h$ is the depth of the tunnel axis; $V_s$ is the formation loss per unit length caused by shield excavation; $R$ is the tunnel excavation diameter; $V_1$ is the formation loss rate caused by shield excavation. In combination with Equations (1) ~ (3), $V_1$ can be obtained as:

$$V_1 = \frac{S_{\max} i \sqrt{2\pi}}{\pi R^2} \tag{4}$$

After normalization of Equation (1), it can be obtained as:

$$\frac{S(x)}{S_{\max}} = \exp\left[-\frac{x^2}{2i^2}\right] \tag{5}$$

Logarithms of both sides of equation (5):

$$\ln\left[\frac{S(x)}{S_{\max}}\right] = -\frac{x^2}{2i^2} \tag{6}$$

From equation (6), it can be seen that the relationship between $\ln[S(x)/S_{\max}]$ and $x^2$ is linear, and the slope is $m = -\frac{1}{2i^2}$. Thus, the width of the settlement tank can be determined as:

$$i = \sqrt{-\frac{1}{2m}} \tag{7}$$

By substituting Equation (7) into (6), $V_1$ can be determined as:

$$V_1 = \frac{S_{\max}\sqrt{-\pi/m}}{\pi R^2} \tag{8}$$

Then, Fig 5 shows the distribution law of surface settlement caused by shield tunnel excavation in strongly weathered rock stratum with different coarse particle content. Fig 6 shows the linear relationship between $\ln[S(x)/S_{\max}]$ and $x^2$ for the formation of strongly weathered rock with different coarse particle contents.

It can be seen from Figs 6(a)–(h) that there is a good linear relationship between $\ln[S(x)/S_{\max}]$ and $x^2$ in each group, and the linear function can be used to fit the data in each group. It can be seen from the fitting results that the correlation coefficient is higher than 0.9, which further indicates that there is a good linear correlation between the data points in each

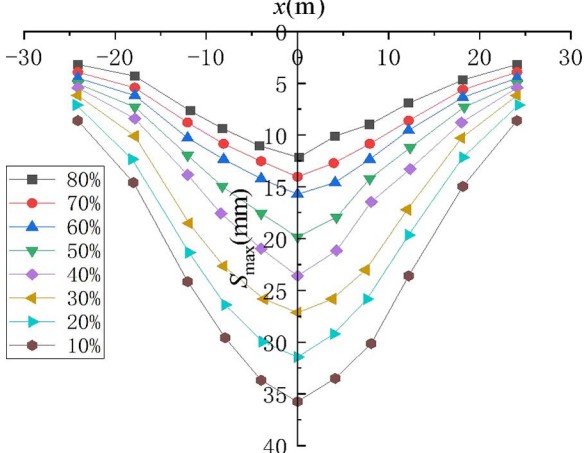

**Fig 5. Surface Settlement Law of Shield Excavation in Strata with Different Coarse Particle Content.**

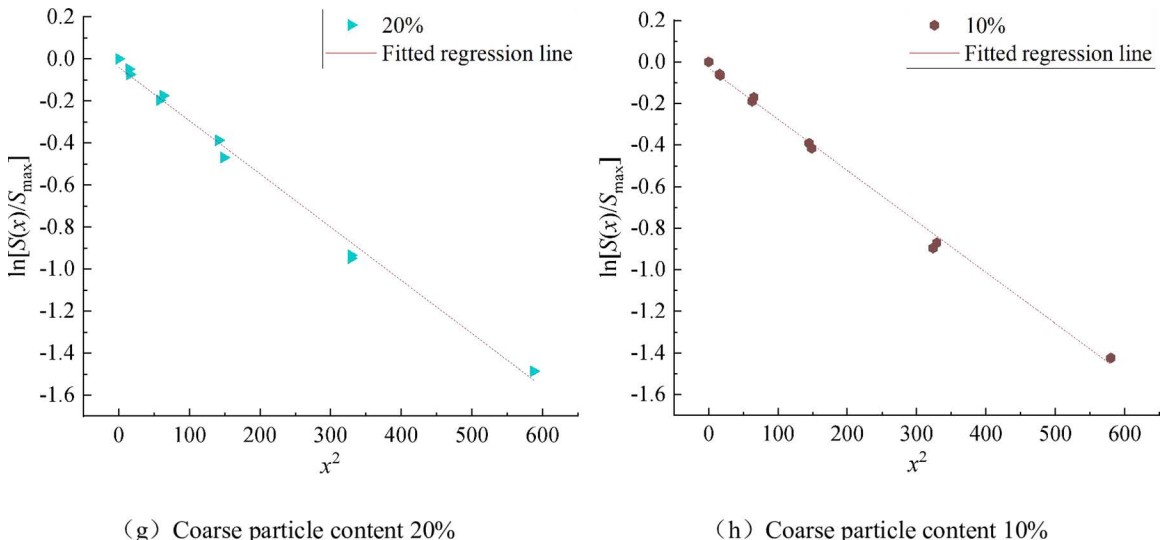

**Fig 6. Linear relationship curve between** $\ln [S(x)/S_{\max}]$ **and** $x^2$ **in strongly weathered rock formations with different coarse particle contents.**

group. The slopes of strongly weathered rock strata with different coarse particle content are shown in Table 4.

It can be seen from Table 4 that with the decrease of coarse particle content, the slope of $\ln [S(x)/S_{\max}] \sim x^2$ fits a straight line of strongly weathered rock stratum and has a trend of gradual increase. According to Equation (8), the loss rate of strongly weathered rocks with different coarse particle contents can be calculated according to the maximum vertical ground settlement above the midpoint of the tunnel. Another important parameter is the excavation radius $R$ of the shield tunnel. The results of the formation loss rate are shown in Table 5.

As can be seen from Table 5, the formation loss rate tends to decrease gradually with the increase of coarse particle content. When the coarse particle content is 10%, the formation loss rate is 12.03%. When the coarse particle content is 80%, the formation loss rate is 4.29%. It shows that more coarse particles can form a space skeleton, improve the bearing deformation capacity of the formation, reduce the loss rate of the formation, and weaken the influence of shield tunnel excavation on the loss of strongly weathered rock formation.

**Table 4. Slope of** $\ln [S(x)/S_{\max}] \sim x^2$ **fitting straight line for strongly weathered rocks with different coarse particle contents.**

| Coarse particle content | 80% | 70% | 60% | 50% | 40% | 30% | 20% | 10% |
|---|---|---|---|---|---|---|---|---|
| $m/10^{-3}$ | -2.23 | -2.14 | -2.13 | -2.29 | -2.41 | -2.56 | -2.53 | -2.45 |

**Table 5. Statistical Table of Formation Loss Rate** $V1$**.**

| Parameter | Coarse particle content | | | | | | | |
|---|---|---|---|---|---|---|---|---|
| | 80% | 70% | 60% | 50% | 40% | 30% | 20% | 10% |
| $m/10^{-3}$ | -2.23 | -2.14 | -2.13 | -2.29 | -2.41 | -2.56 | -2.53 | -2.45 |
| $S_{\max}$ / mm | 12.17 | 14.04 | 15.72 | 19.84 | 23.58 | 27.14 | 31.44 | 35.75 |
| $R$/ m | 5.82 | 5.82 | 5.82 | 5.82 | 5.82 | 5.82 | 5.82 | 5.82 |
| $V_1$ / % | 4.29 | 5.06 | 5.68 | 6.91 | 8.00 | 8.94 | 10.42 | 12.03 |

# 4 Investigation of surface deformation in shield construction through strongly weathered rock formation with coarse particle content

## 4.1 Peck formula principle introduction

Peck [20] analyzed a large number of measured surface data of tunnel engineering and found that the surface settlement was approximately normally distributed horizontally. Assuming that the volume of ground loss caused by tunnel excavation is equal to the volume of the surface settlement tank, the ground settlement law can be described by the Gaussian normal distribution curve. Then, the Peck settlement tank equation is obtained as follows [20]:

$$S(x) = S_{\max} \exp\left(-\frac{x^2}{2i^2}\right)$$

(9)

$$S_{\max} = \frac{V_l}{i\sqrt{2\pi}}$$

(10)

$$i = \frac{Z}{\sqrt{2\pi}\tan(45° - \phi/2)}$$

(11)

where $S(x)$ is the ground settlement from the middle line $x$ of the tunnel; $S_{\max}$ is the ground settlement at the middle line of the tunnel; $x$ is the distance from the tunnel center line; $i$ is the width of settlement tank, which is the horizontal coordinate of the reverse bending point of sedimentation curve; $V_l$ is the volume lost per unit length of tunnel caused by shield construction; $\varphi$ is the friction angle in the formation around the tunnel; $Z$ is the depth from the surface to the center of the tunnel.

The next step provides an introduction to the basic principles of random medium, which facilitates subsequent comparative analyses.

## 4.2 Basic principles of random medium theory

The random medium theory equates the influence of the whole tunnel excavation to the sum of the influence of the infinitesimal excavation elements on the surface. For a single-hole shield tunnel excavated at a certain depth, the surface settlement caused by excavation can be regarded as a plane strain problem, and the element excavation is infinite along the Y-axis. After a long time, the surface subsidence caused by the element excavation reaches the maximum value, and the final settlement is expressed by $We(x)$.

$$We(x) = \frac{1}{r(z)} \exp\left[-\frac{\pi}{r(z)^2}x^2\right] d\xi d\eta$$

(12)

where, r(z) is the main influence range of the unit excavation at the z level. The formation main influence angle β is considered, and it is considered that r(z) has a linear relationship with depth z:

$$r(z) = \frac{z}{\tan\beta}$$

(13)

where, $\tan\beta$ depends on the ground condition where the excavation is taking place. For the surface, the main influence range $r(H) = H/\tan\beta$.

Assuming that every excavation unit in the entire $\Omega$ excavation range collapses completely, the superposition principle is adopted and (13) is substituted into (12) to obtain the surface settlement:

$$W_{\Omega}(x) = \int_a^b \int_c^d \frac{\tan\beta}{\eta} \exp\left[-\frac{\pi\tan^2\beta}{\eta^2}(x-\xi)^2\right] d\xi d\eta \tag{14}$$

In the process of tunnel shield excavation, due to the cutterhead support pressure and shield shell support force, the shield tail will collapse slightly due to grouting and segment support, resulting in formation loss and then cause formation loosening and deformation. In the process of shield construction, soil mass at the excavation boundary of shield tunneling did not shrink uniformly. Loganathan et al. [29] proposed a convergence model of non-uniform radial soil mass and calculated it on the basis of Verruijt [30] and Booker's [31] general Sagaseta solution. The movement mode of soil mass around the tunnel plane was shown in Fig 7.

According to the superposition principle, it can be seen that the surface settlement tank volume is equal to the settlement difference caused by the excavation range $\Omega$ and $\omega$, namely:

$$W(x) = W_{\Omega}(x) - W_{\omega}(x) \tag{15}$$

$$W_{\omega} = \int_e^f \int_g^h \frac{\tan\beta}{\eta} \exp\left[-\frac{\pi\tan^2\beta}{\eta^2}(x-\xi)^2\right] d\xi d\eta \tag{16}$$

Also according to the superposition principle, the horizontal displacement U(x) of each point on the surface of the random medium theory is as follows:

$$U(x) = \int\int_{\Omega-\omega} \frac{(x-\xi)\tan\beta}{\eta} \exp\left[-\frac{\pi\tan^2\beta}{\eta^2}(x-\xi)^2\right] d\xi d\eta \tag{17}$$

Calculation of the upper and lower limits of the differentiation in the above equation: $a = (H-R), b = H+R$ ; $c = -\sqrt{R^2-(H-\eta)^2}, d = \sqrt{R^2-(H-\eta)^2}$; $e = (H-R+g), f = H+R$; $g = -\sqrt{(R-0.5g)^2-(H+0.5g-\eta)^2}, h = \sqrt{(R-0.5g)^2-(H+0.5g-\eta)^2}$.

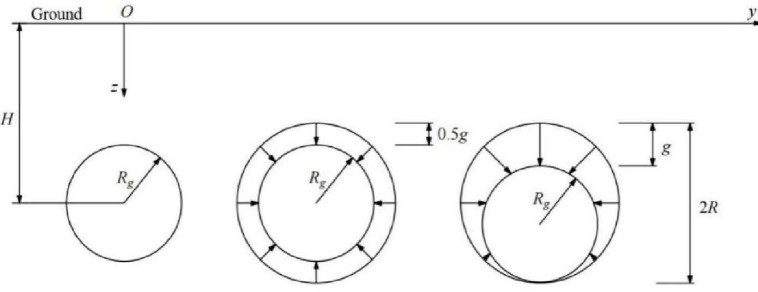

**Fig 7. Movement pattern of the soil around the tunnel plane.**

Two important parameters should be determined by random medium theory in the prediction of surface settlement in shield excavation: the formation influence angle β (reflecting the width and depth of the settlement tank) and the soil loss parameter g (reflecting the depth of the settlement tank).

In the process of shield tunneling, there will be a certain gap between the pipe segment behind the shield tunneling machine and the surrounding soil layer, which will cause formation loss and then cause formation loosening and deformation. Sedimentation tanks in strata with different burial depths are different, so it is necessary to calculate sedimentation tanks with different burial depths. The calculation equation is as follows [23]:

$$\tan\beta = \frac{H-z}{i_z\sqrt{2\pi}}$$

(18)

$$i_z = i_0 \cdot \left(1-\frac{z}{H}\right)^{0.3}$$

(19)

$$i_0 = k \cdot H$$

(20)

$$k = 1 - 0.02\varphi$$

(21)

where, $i_z$ is the width of the soil settlement tank at the underground depth z, $i_0$ is the width of the surface soil settlement tank, $H$ is the buried depth of the tunnel axis, $k$ is the width coefficient of the surface settlement tank, and $\varphi$ is the formation friction angle.

From Equations (18) to (21), it can be obtained:

$$\tan\beta = \frac{1}{k \cdot \left(\frac{1-a(z/H)}{1-z/H}\right)\sqrt{2\pi}}$$

(22)

By substituting Equation (22) into Equations (16) and (17), the vertical displacement and horizontal deformation of the soil layer at depth $z$ can be obtained.

## 4.3 Extended analysis of stochastic medium theory

For Peck equation method, the previous analysis shows that:

$$S(X) = \frac{V_1}{\sqrt{2\pi}i}exp(-\frac{X^2}{2i^2})$$

(23)

Due to the assumption that the volume does not deform in both Peck equation and random medium theory, the total volume of surface settlement is equal to the volume reduction caused by underground tunnel excavation. Then the formation loss $V_1$ is written as an integral, that is $V_l = \iint\limits_{\Omega} d\xi d\eta$, the Peck equation can be written as:

$$S(X) = \iint \frac{1}{\sqrt{2}}exp(-\frac{X^2}{2i^2})d\xi d\eta$$

(24)

Compare the vertical settlement formula in the theory of random media with it:

$$W(X) = \int\int \frac{1}{r(Z)} exp\left[-\frac{\pi}{r^2(Z)}X^2\right] d\xi d\eta \tag{25}$$

In contrast, when $r(Z) = \sqrt{2\pi}i$, the two equations are exactly the same. This conclusion has also been demonstrated in literature [32]. As discussed above, $r(Z)$ represents the radius of the influence range of the surface, and $i$ represents the horizontal distance from the reverse bend point on the settlement curve to the center line of the tunnel, and there is a linear relationship between the two equations.

After a comparative analysis of the two equations, it is found that the settlement curve obtained by the random medium theory is the integral superposition of the Peck Gaussian curve, and the Peck equation is a special case in which the excavation section in the random medium theory is regarded as an infinitely small circular element. Because the Peck equation is based on the circular tunnel summary, it is a simplified form of random medium theory which is less applicable, and its settlement curve is relatively simplified. Thus, the random medium theory has a wider application range and can be applied to tunnels of any cross-section form, and the calculated results are more reliable [25]. It can be inferred that when the actual tunnel section is small enough relative to its buried depth, the two results should be approximately consistent.

Theoretically, the calculated results of random medium theory are closer to the actual form, but it is difficult to obtain the cross-section deformation form and the formation main influence angle β in random medium theory. For the deformation form of the most common circular tunnel section in practical engineering, after simplifying the two cases of uniform convergence and non-uniform convergence, the unknown parameters in the random medium theoretical formula are only $\Delta A$ (representing the convergence of the tunnel section) and $\tan\beta$ (representing the tangent value of the main influence angle of excavation). According to the relationship between the random medium theory established above and the Peck settling tank equation, the relationship between $\Delta A$, $\tan\beta$, and the parameters in the Peck equation can be obtained.

For a circular tunnel with radius A, the formation loss rate $V_l$ in Peck's equation can be written as:

$$V_1 = \frac{\pi A^2 - \pi(A - \Delta A)^2}{\pi A^2} \tag{26}$$

and,

$$\Delta A = A(1 - \sqrt{1 - V_l}) \tag{27}$$

In the relation between the established random medium theory and Peck's equation, $r(Z) = \sqrt{2\pi}i$, then

$$r(Z) = \frac{\eta}{\tan\beta} = \sqrt{2\pi} \cdot i \tag{28}$$

Setting the mean of the circular tunnel to be η, the distance H from the center of the tunnel to the surface is:

$$\tan\beta = \frac{H}{\sqrt{2\pi} \cdot i} \tag{29}$$

At present, the most widely used method for determining the width of the settlement tank is the empirical formula obtained by O 'Reilly and New on the basis of engineering-measured data, that is, $i = KZ_0$ [33], and then:

$$\tan \beta = \frac{H}{\sqrt{2\pi} \cdot i} = \frac{1}{\sqrt{2\pi K}}$$

(30)

From equation (30), it can be seen that the width coefficient $K$ of the sedimentation tank is negatively correlated with the angle of β. Based on the Peck parameter characteristics, the following conclusions can be drawn: the harder the soil layer, the larger the internal friction angle φ, the smaller the sedimentation trough width coefficient $K$, the larger the formation main influence angle β, and the smaller the horizontal influence range of the formation. According to equations (27) and (28), the unknown parameters $\Delta A$ and tanβ in the random medium theory can be calculated by the values of parameters $V_l$ and $K$ in Peck's equation. A large number of Peck empirical parameters are converted into random medium theoretical parameters, which greatly expands the application range of random medium theory in engineering.

The conversion relationship between parameters $V_l$ and $K$ in the Peck equation and unknown parameters $\Delta A$ and tanβ in random medium theory have been derived above. However, the formation loss rate and subsidence trough width of strongly weathered rock stratum caused by shield construction are not clear, so $V_l$ and $K$ values in the Peck equation can be calculated by combining them with the relationship between particle size and formation loss. Then, from the linear regression analysis of the Peck equation:

$$\ln S(x) = \ln S_{\max} + \frac{1}{i^2} \left( -\frac{x^2}{2} \right)$$

(31)

Take $\ln S(x)$ and -x^2/2 as the regression variable; set $\ln S(x)$ as the constant term after regression, and -x^2/2 as the linear coefficient after regression. The regression process is as follows:

$$S_{xx} = \sum \left( -\frac{x_i^2}{2} \right)^2 - \frac{1}{n} \left( \sum \frac{x_i^2}{2} \right)^2$$

(32)

$$S_{xy} = \sum \left[ \left( -\frac{x_i^2}{2} \right) \ln S(x_i) \right] - \frac{1}{n} \sum \left( -\frac{x_i^2}{2} \right) \sum \ln S(x_i)$$

(33)

$$S_{yy} = \sum \ln^2 S(x_i) - \frac{1}{n} \left[ \sum \ln S(x_i) \right]^2$$

(34)

$$\hat{b} = \frac{S_{xy}}{S_{xx}}$$

(35)

$$\hat{a} = \overline{\ln S_i(x)} - \hat{b} \times \overline{(-x^2)/2}$$

(36)

The linear regression equation becomes

$$\ln S(x) = \hat{a} + \hat{b}\left(-\frac{x^2}{2}\right) \tag{37}$$

where $\hat{a}$ is the constant term of the regression equation; $\hat{b}$ is the linear coefficient of the regression equation; $x_i$ is the distance from the $i_{th}$ settlement monitoring point to the tunnel center line.

Then,

$$S_{\max} = \exp(\hat{a}) \tag{38}$$

$$i = 1/\sqrt{\hat{b}} \tag{39}$$

The regression curve is

$$S = \exp(\hat{a})\exp\left(-\frac{\hat{b}x^2}{2}\right) \tag{40}$$

The linear correlation was tested by $R$.

$$R = \frac{S_{xy}}{\sqrt{S_{xx}}\sqrt{S_{yy}}} \tag{41}$$

When the condition $r_{0.01}(n-2) > R > r_{0.05}(n-2)$ is satisfied, the linear relationship of the regression function is significant.

Fitting of the sedimentation curves to the monitoring data of different coarse particle contents in the above text, due to the repetition of the calculation process, Table 6 is used here as an example, and the fitting process for the rest of the coarse particle contents is omitted.

Taking the data in Table 6 as an example, it can be obtained by using equations (32) ~ (34) that $S_{xx} = 15661.93$, $S_{xy} = 751.75$, $S_{yy} = 3.61$. Calculating through equations (35) and (36):

**Table 6. Regression Analysis of 80% Coarse Particle Content Sedimentation Data.**

| Sample point | Abscissa (x)/ m | S(x)/ mm | $\frac{-x^2}{2}$ | $\ln|S(x)|$ |
|---|---|---|---|---|
| 1 | 25 | -2.24 | 312.5 | 0.81 |
| 2 | 20 | -3.85 | -200 | 1.35 |
| 3 | 15 | -5.85 | 112.5 | 1.77 |
| 4 | 10 | -7.81 | -50 | 2.06 |
| 5 | 6 | -9.14 | -18 | 2.21 |
| 6 | 3 | -9.98 | -4.5 | 2.30 |
| 7 | 0 | -10.13 | 0 | 2.32 |
| 8 | 3 | -9.85 | -4.5 | 2.29 |
| 9 | 6 | -9.30 | -18 | 2.23 |
| 10 | -10 | -7.99 | -50 | 2.08 |
| 11 | -15 | -5.93 | 112.5 | 1.78 |
| 12 | -20 | -3.88 | -200 | 1.36 |
| 13 | -25 | -2.26 | 312.5 | 0.82 |

$$\hat{a} = 2.311, \hat{b} = 0.00480 \tag{42}$$

The fitted functions obtained for each coarse particle content are shown in Table 7. By comparing the monitored data of various coarse particle content conditions with the regression function, the results shown in Fig 8 are obtained.

As can be seen from Fig 8, the linear function after regression fits well with the measured data, and the measured data are evenly distributed around the linear function after regression, which can better represent the relationship between the two. According to the above analysis, the revised Peck equation is:

$$S(x) = \alpha S_{\max} \exp\left(\frac{-x^2}{2(\beta i)^2}\right) \tag{43}$$

Where $\alpha$ is the correction coefficient of the maximum surface settlement; $\beta$ is the correction factor for the width of the settlement tank; $S_{max}$ is the maximum surface settlement in the Peck equation.

Then, after a linear transformation of Equation (43):

**Table 7. Fitting function for each coarse particle content.**

| Number | Coarse particle content | Fitting function |
|---|---|---|
| 1 | 80% | $\ln S(x) = 2.311 + 0.00480 \times \left(\frac{-x^2}{2}\right)$ |
| 2 | 70% | $\ln S(x) = 2.530 + 0.00520 \times \left(\frac{-x^2}{2}\right)$ |
| 3 | 60% | $\ln S(x) = 2.731 + 0.00569 \times \left(\frac{-x^2}{2}\right)$ |
| 4 | 50% | $\ln S(x) = 2.914 + 0.00627 \times \left(\frac{-x^2}{2}\right)$ |
| 5 | 40% | $\ln S(x) = 3.075 + 0.00684 \times \left(\frac{-x^2}{2}\right)$ |
| 6 | 30% | $\ln S(x) = 3.230 + 0.00755 \times \left(\frac{-x^2}{2}\right)$ |
| 7 | 20% | $\ln S(x) = 3.379 + 0.00838 \times \left(\frac{-x^2}{2}\right)$ |
| 8 | 10% | $\ln S(x) = 3.522 + 0.00940 \times \left(\frac{-x^2}{2}\right)$ |

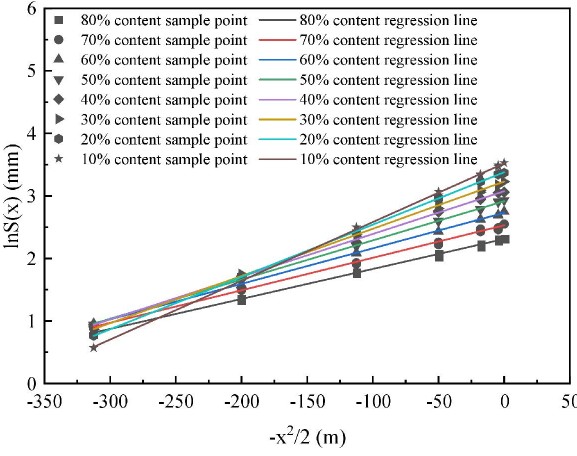

**Fig 8. Regression situation of each cross-section.**

$$\ln S\left(x\right) = \ln \alpha S_{\mathrm{max}} + \left(\frac{-x^2}{2\left(\beta i\right)^2}\right)$$

(44)

Set $\ln \alpha S_{\mathrm{max}}$ as a constant term and $1/\left(\beta i\right)^2$ as a linear coefficient, and then substitute the values of $\hat{a}$ and $\hat{b}$ calculated by Equation (42) into Equation (44):

$$\alpha = \frac{\exp \left(\hat{a}\right)}{S_{\mathrm{max}}}$$

(45)

$$\beta = \frac{1}{i\left(\hat{b}\right)^{0.5}}$$

(46)

$S_{\mathrm{max}}$ and $i$ after regression can be calculated using the obtained $\hat{a}$、$\hat{b}$, and then using the fitted curve, the results of parameter α and β values of the eight working conditions can be fitted, as shown in Fig 9.

The fitting equation of the maximum correction factor for surface settlement $a$ to the coarse particle content during shield excavation in strongly weathered rock strata is shown in Eq. (47), and the equation's coefficient of determination $R^2 = 0.999$.

$$y = 0.807 + 0.0192x + 1.617x^2$$

(47)

The fitting equation for the settlement tank width correction factor $\beta$ with respect to the coarse particle content is shown in equation (48), with a coefficient of determination $R^2 = 0.999$.

$$y = 5.001 - 0.0196x$$

(48)

According to the engineering geology, the correction coefficients $a$ and $\beta$ of Peck's formula considering the coarse particle content can be derived by bringing the coarse particle content in the strongly weathered rock layer into Eqs. (47) and (48), and in further, using Eq. (43), the

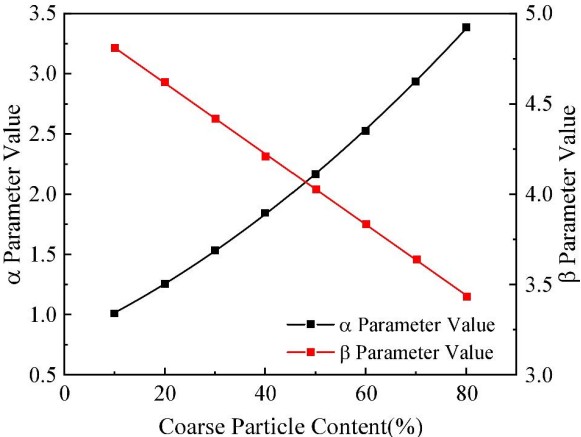

**Fig 9. Fitting Results for Each Operating Condition.**

surface settlement caused by the shield construction process can be predicted for the strongly weathered rock layer considering the coarse particle content.

## 5 Verification and application of formation deformation prediction method

Through the shield tunnel project of the Pazhou Branch line in the Pearl River Delta, the revised theoretical equation is adopted to analyze the construction impact of curve excavation of shield machine in strongly weathered rock, solve the surface deformation caused by shield tunnel excavation, and compare the settlement value calculated by the revised equation with the field measured value, numerical simulation value, and original theoretical equation value. The validity and reliability of the calculation method proposed in this study are verified.

The total length of the shield tunnel of Pazhou Branch line in the Pearl River Delta Intercity is 3257.36m, with an excavated diameter of 9.14m, a length of 7.1m, an outer diameter of the segment of 8.8m, an inner diameter of 8.0m, the ring width 1.8m, and the thickness 0.4m. The shield tunnel roof is covered with soil 2.98 ~ 34.45m. The maximum slope of the tunnel is 30‰, and the minimum curve radius is 800m. The traversing strata are mainly mixed strata of silty clay and weathered granite, among which weathered granite strata are fully weathered, strongly weathered, and partly moderately weathered granite stratas. The average pressure of the shield bin is maintained at 240 ~ 300kPa, which is slightly higher than the static earth pressure at the center of the tunnel, so as to cause a slight uplift of the square soil layer before excavation, which is used to compensate for the soil loss caused by the ejection of the shield tail. In this study, a more typical section in the construction section is selected for analysis,

Table 8.  Typical Cross Section Calculation Parameters of Shield Tunnel on Pazhou Branch Line.

| Cross Section name | Tunnel depth/ m | Coarse particle content | Width coefficient of sedimentation tank | Strata loss ratio |
|---|---|---|---|---|
| Cross Section 1 | 15.2 | 37% | 4.0 | 1.05% |
| Cross Section 2 | -19.4 | 53% | 5.2 | 0.97% |

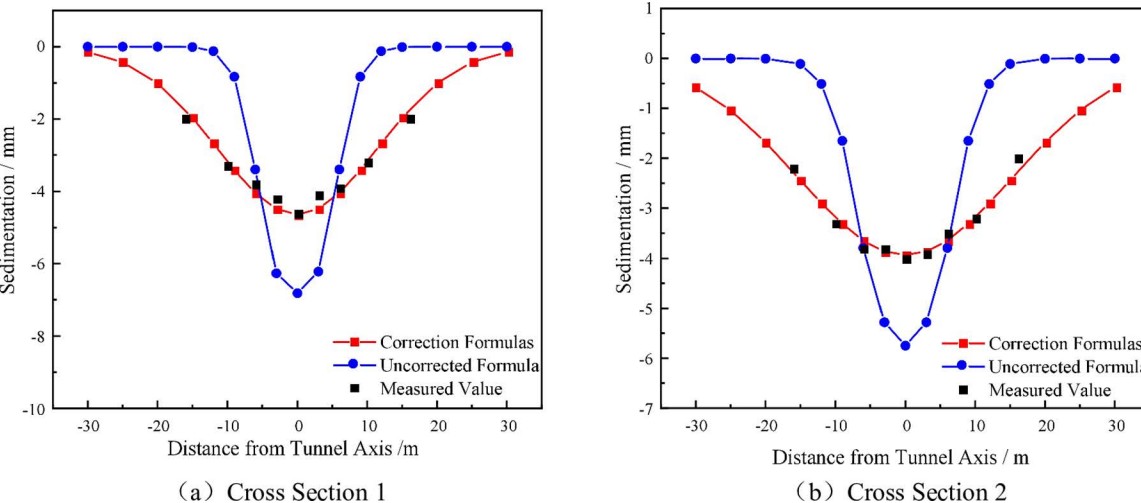

（a）Cross Section 1　　　　（b）Cross Section 2

**Fig 10.  Comparison between Calculated and Measured Values of Typical Sections.**

and its measured values and calculated values are compared and analyzed respectively. The relevant calculation parameters of the section are shown in Table 8.

Fig 10 shows the comparison between calculated and measured values of typical sections. By comparing the calculated values of the random medium theory and the modified random medium theory equations with the measured values respectively, it can be found that the final settlement curve calculated by the method considering the change of coarse particle content in the strongly weathered rock formation is closer to the measured values, and the formation settlement can be better predicted by this method.

To sum up, the calculation method in this study better reflects the surface deformation law caused by shield tunneling machine construction in strongly weathered rock formations. Compared with traditional analytical methods and numerical simulation, it can better reflect the comprehensive impact of changes in coarse particle content on various construction parameters during construction, which can provide a theoretical basis for shield tunneling construction in strongly weathered rock formations

## 6  Conclusions

This study is based on the Guangzhou-Zhuhai Intercity Railway Pazhou Branch Line tunnel project in the Pearl River Delta, where a method to predict surface settlement caused by subway tunnel shield tunneling is applied. The Peck formula is used for linear regression analysis to further explore how the content of coarse particles in strongly weathered rock layers influences surface settlement due to shield tunneling. The results indicate that the physical and mechanical properties of the strongly weathered rock layers have a significant impact on tunneling stability, especially the variations in coarse particle content, which directly affect surface settlement and construction safety. Through physical and mechanical testing of the strongly weathered rock samples, it is found that as the coarse particle content increases, both the dry density and compression coefficient decrease, while the compression modulus increases. Additionally, the load transfer mechanism shifts from compression deformation of fine-grained soil to point contact transfer between coarse particles. Therefore, it is essential to adjust the construction parameters in real-time based on the content of coarse particles to ensure safety and quality during tunnel construction.

Furthermore, the study shows that the coarse particle content significantly influences the stratum loss rate. As the content of coarse particles increases, the stratum loss rate gradually decreases. For instance, when the coarse particle content is 10%, the stratum loss rate is 12.03%, while at 80%, it drops to 4.29%. This demonstrates that coarse particles form a stable skeleton in the soil, increasing the bearing capacity of the strata and reducing the damage to the soil layer caused by tunnel construction. Based on this finding, the study suggests that special attention should be paid to the distribution and content of coarse particles during shield tunneling in strongly weathered rock layers, in order to minimize stratum loss and uneven settlement during construction.

Finally, by combining the Peck empirical formula with the random media theory, the study improves the surface settlement prediction model. Linear regression analysis shows that the modified Peck formula can accurately predict the surface settlement caused by shield tunneling in strongly weathered rock layers. By introducing correction parameters $\alpha$ and $\beta$, the study provides a reliable method for predicting settlement in this region, with high practical value. In the actual engineering project of the Guangzhou-Zhuhai Intercity Railway Pazhou Branch Line, the study's results show good agreement with measured data, proving that the model is both feasible and accurate for guiding shield tunnel construction in similar geological conditions.

## Author contributions

**Data curation:** Bao-xin Jia.

**Formal analysis:** Qinglei Yuan.

**Investigation:** Puzhen An, Lei Chen.

**Writing – original draft:** Bao-xin Jia.

**Writing – review & editing:** Bao-xin Jia.

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
