## [Decision Letter · Decision Letter 0]

11 Nov 2024

PONE-D-24-45489Study on surface deformation caused by shield tunnel excavation caused by coarse particle content in strongly weathered rock formationsPLOS ONE

Dear Dr. Chen,

Thank you for submitting your manuscript to PLOS ONE. After careful consideration, we feel that it has merit but does not fully meet PLOS ONE’s publication criteria as it currently stands. Therefore, we invite you to submit a revised version of the manuscript that addresses the points raised during the review process.

**ACADEMIC EDITOR: **

The review process is now completed and we have three reports submitted by anonymous reviewers. As can be seen from their reports, all reviewers request revision beforethis manuscript is reviewed again. Thus, I invite you for a substantial revision.

If you have any questions or need further clarification on the revisions required, please do not hesitate to contact me. I am here to assist you throughout this process.

Thank you for your attention to this matter, and I look forward to receiving your revised manuscript.

Meanwhile, I want to thank the reviewers for their great efforts on your manuscript and you for your submission.

We look forward to receiving your revised manuscript.

Kind regards,

Linwei Li

Academic Editor

PLOS ONE

Journal Requirements:

2. Please provide additional details about the origin of samples used in this study including location.

4. Thank you for stating the following in the Acknowledgments Section of your manuscript: [Te research was supported by the Funded by Discipline Innovation team of Liaoning Technical University (No.LNTU20TD08). We appreciate the editor and anonymous reviewers’ useful comments.]

Please remove any funding-related text from the manuscript and let us know how you would like to update your Funding Statement. Currently, your Funding Statement reads as follows: [The authors received no specific funding for this work]

5. We note that your Data Availability Statement is currently as follows: [All relevant data are within the manuscript and its Supporting Information files.]

6. PLOS requires an ORCID iD for the corresponding author in Editorial Manager on papers submitted after December 6th, 2016. Please ensure that you have an ORCID iD and that it is validated in Editorial Manager. To do this, go to ‘Update my Information’ (in the upper left-hand corner of the main menu), and click on the Fetch/Validate link next to the ORCID field. This will take you to the ORCID site and allow you to create a new iD or authenticate a pre-existing iD in Editorial Manager.

Reviewers' comments:

Reviewer's Responses to Questions

**Comments to the Author**

1. Is the manuscript technically sound, and do the data support the conclusions?

Reviewer #1: Yes

Reviewer #2: Partly

Reviewer #3: Partly

2. Has the statistical analysis been performed appropriately and rigorously? 

Reviewer #1: Yes

Reviewer #2: Yes

Reviewer #3: Yes

3. Have the authors made all data underlying the findings in their manuscript fully available?

Reviewer #1: Yes

Reviewer #2: Yes

Reviewer #3: Yes

4. Is the manuscript presented in an intelligible fashion and written in standard English?

Reviewer #1: Yes

Reviewer #2: Yes

Reviewer #3: Yes

5. Review Comments to the Author

Reviewer #1: Taking Pazhou Branch shield tunnel project in Pearl River Delta as an example, this paper analyzes the influence of coarse grain content in heavily weathered rock on the mechanical properties of heavily weathered rock, and studies the influence of coarse grain content on the surface loss caused by tunnel shield excavation. By comparing Peck formula with random medium theory, a large number of Peck empirical parameters are transformed into random medium theory parameters, which expands the application range of random medium theory in engineering. At the same time, considering the coarse particle content in the heavily weathered rock, the Peck formula was analyzed by linear regression, and the maximum settlement correction coefficient and subsidence width correction coefficient β were introduced to modify the Peck formula. This is an interesting paper, logically rigorous and well-organized, which should be considered for publication, but the following minor revisions should be made before publication:

1. Although the paper includes curves, the horizontal and vertical axes should be labeled according to the journal's template.

2. The hysteresis loop in Figure 4 is not distinct, so it is recommended to enlarge it for clarity.

3. All tables in this paper should be formatted according to the journal's template.

4. It is recommended to use a formula editor when writing formulas, and ensure that variables and symbols within the formulas are italicized or not.

5. As observed in Figure 10, the revised theoretical calculation results presented in this paper are significantly higher than the unrevised results, indicating a large error. Please further verify the calculations.

6. The conclusion section is overly verbose, so it is suggested to condense it into three paragraphs.

Reviewer #2: The manuscript titled "Study on surface deformation caused by shield tunnel excavation caused by coarse particle content in strongly weathered rock formations" contributes to the understanding of how geological composition, specifically coarse particle content, affects tunneling-induced surface deformation and provides improved tools for predicting these effects in strongly weathered rock formations. The content is interesting and can contribute to the rock engineering. However, the current version of this manuscript should be improved for publication. Please see my comments below:

1. First, all the formats should be consistent. The font should consistent and make it more readability, especially for the subtitle. The font in the figures should be enlarged and the fonts should be consistent with the text of the manuscript. The table titles should move above the table. The authors should make the figures clearer.

2. The title is repetitive with "caused by" appearing twice. Please revise the title.

3. In the Abstract section, the abstract is quite long and please reduce it for better readability.

4. “At the same time, the linear regression analysis of Peck's formula is carried out by considering the content of coarse particles in the strongly weathered rock layer, and the regressed linear function is fitted to the measured data with different coarse particles content, and the maximum surface The maximum subsidence correction coefficient � and the sinkhole width correction coefficient β were introduced to correct the formula, and finally the reliability of the formula derived in this paper was verified.” This sentence in the abstract is too long and is awkwardly phrased. Please revise it for clarity.

5. In the Introduction, the literature review is comprehensive, but it could benefit from a clearer structure. And the research gap could be more explicitly stated at the end of the introduction. Additionally, the outline of the manuscript also should be added at the end of the introduction.

6. The literature review appears to be somewhat limited in scope. Including more diverse studies could provide a more comprehensive background and better context for your research. For example: Combined effects of fault geometry and roadway cross-section shape on the collapse behaviors of twin roadways: An experimental investigation; Risk assessment of ground collapse along tunnels in karst terrain by using an improved extension evaluation method.

7. The methodology is not clear introduced in the manuscript for replicability.

8. For the equations, please ensure all variables are defined immediately after each equation.

9. The transition between the Peck formula and random medium theory could be smoother. Consider adding a brief paragraph explaining why both methods are being used.

10. Explain why only two cross-sections were chosen for analysis. Are these representative of the entire tunnel?

11. There are occasional grammatical errors throughout the manuscript. For example, "Te research was supported by..." in the Acknowledgements should be "The research was supported by..." . Some sentences are overly long and complex. Consider breaking them into shorter, clearer statements.

12. For the references, ensure all DOIs are correctly formatted and active and the format should be consistent.

Reviewer #3: (1) The title is not fluent, and its meaning is unclear.

(2) Abstract: The word count is excessive, and the content is overly detailed, requiring condensation. The expression needs refinement, and phrases such as "the author" should be avoided.

(3) The sections of the paper are not numbered, making it appear disorganized and lacking in structure and logical flow. Additionally, the formatting of the equations is inconsistent, contributing to an overall lack of cohesion.

(4) The entire paper requires language refinement, as the expressions are inconsistent and unclear. Furthermore, the font style should be standardized, as exemplified by the inconsistent use of terms such as "strongly weathered rock" and "heavily weathered rock."

(5) “Research on a mathematical model of surface deformation of heavily weathered rock stratum in shield excavation”, the title of this section needs to be revised to better highlight the focus on the development and validation of the computational model.

(6) The structure of this paper is more like a thesis, as it is divided into three sections: the impact of coarse particle content on the mechanical properties of highly weathered rock layers, the effect of shield tunneling on ground loss, and the proposal of a computational model. However, these sections are disconnected and lack cohesion, making the paper feel disorganized and overly lengthy.

(7) Figure 3 and 6 require enhancement in terms of presentation. Some annotations in the figures are too small and unclear, and any unnecessary content should be removed from the charts.

(8) Tables 6 to 13 illustrate the calculation process for the fitted curves, comprising almost 8 pages of the manuscript. This extensive presentation is redundant and could be significantly condensed.

(9) How is the consistency between the fitted curve and the measured data reflected in Figure 9? Moreover, there is an inconsistency in the description of the settlement tank width in this section, a discrepancy that appears multiple times throughout the manuscript. This highlights the need for greater precision and consistency in the expression used within the paper.

(10) It’s good to cite these related papers in the introduction and to improve your expression referring to these papers

Soft computing approach for prediction of surface settlement induced by earth pressure balance shield tunneling

https://doi.org/10.1016/j.undsp.2019.12.003

Investigating Surface Deformation Caused by Excavation of Curved Shield in Upper Soft and Lower Hard Soil

https://doi.org/10.3389/feart.2022.844969

Ground and tunnel responses induced by partial leakage in saturated clay with anisotropic permeability

https://doi.org/10.1016/j.enggeo.2015.02.005

Seepage propagation simulation of a tunnel gasketed joint using the cohesive zone model

https://doi.org/10.1016/j.tust.2024.105726

Time-varying compressive properties and constitutive model of EPDM rubber materials for tunnel gasketed joint

https://doi.org/10.1016/j.conbuildmat.2024.136734

6. PLOS authors have the option to publish the peer review history of their article (what does this mean?). If published, this will include your full peer review and any attached files.

Reviewer #1: No

Reviewer #2: No

Reviewer #3: No

---

## [Author Response · Author response to Decision Letter 1]

7 Jan 2025

Dear professor Linwei Li editor and reviewers,

We have addressed each comment carefully and made corrections as needed. The revised portions are marked in red in the paper.

---

## [Decision Letter · Decision Letter 1]

18 Feb 2025

PONE-D-24-45489R1Study on surface deformation induced by shield excavation due to coarse particle content in strongly weathered rock layersPLOS ONE

Dear Dr. Jia,

Thank you for submitting your manuscript to PLOS ONE. After careful consideration, we feel that it has merit but does not fully meet PLOS ONE’s publication criteria as it currently stands. Therefore, we invite you to submit a revised version of the manuscript that addresses the points raised during the review process.

We look forward to receiving your revised manuscript.

Kind regards,

Linwei Li

Academic Editor

PLOS ONE

Comments from the Editorial Office on behalf of the Academic Editor:

We note that Reviewer 3 recommended that you cite specific previously published works in an earlier round of revision. As always, we recommend that you please review and evaluate the requested works to determine whether they are relevant and should be cited. It is not a requirement to cite these works and you may remove them before the manuscript proceeds to publication. We appreciate your attention to this request.

Reviewers' comments:

Reviewer's Responses to Questions

**Comments to the Author**

1. If the authors have adequately addressed your comments raised in a previous round of review and you feel that this manuscript is now acceptable for publication, you may indicate that here to bypass the “Comments to the Author” section, enter your conflict of interest statement in the “Confidential to Editor” section, and submit your "Accept" recommendation.

Reviewer #1: All comments have been addressed

Reviewer #2: All comments have been addressed

Reviewer #3: All comments have been addressed

2. Is the manuscript technically sound, and do the data support the conclusions?

Reviewer #1: Yes

Reviewer #2: Yes

Reviewer #3: Yes

3. Has the statistical analysis been performed appropriately and rigorously? 

Reviewer #1: Yes

Reviewer #2: Yes

Reviewer #3: Yes

4. Have the authors made all data underlying the findings in their manuscript fully available?

Reviewer #1: Yes

Reviewer #2: Yes

Reviewer #3: Yes

5. Is the manuscript presented in an intelligible fashion and written in standard English?

Reviewer #1: Yes

Reviewer #2: Yes

Reviewer #3: Yes

6. Review Comments to the Author

Reviewer #1: The responses to the comments are sufficient. The improvement of the manuscript is significant. Please consider accepting as is.

Reviewer #2: As the authors have addressed all the comments raised by the reviewers, I have no further comments on this manuscript.

Reviewer #3: (No Response)

7. PLOS authors have the option to publish the peer review history of their article (what does this mean?). If published, this will include your full peer review and any attached files.

Reviewer #1: No

Reviewer #2: No

Reviewer #3: No

---

## [Author Response · Author response to Decision Letter 2]

21 Feb 2025

We have addressed each comment carefully and made corrections as needed. The revised portions are marked in red in the paper. The response to the editorial follows:

(1) We note that Reviewer 3 recommended that you cite specific previously published works in an earlier round of revision. As always, we recommend that you please review and evaluate the requested works to determine whether they are relevant and should be cited. It is not a requirement to cite these works and you may remove them before the manuscript proceeds to publication. We appreciate your attention to this request.

Response: Thanks to your reminder, I have scrutinized the references in the article, reviewing and evaluating the references cited and removing those of low relevance to the topic of the article.

Response: Thank you for the correction, I have double checked the reference list for completeness and accuracy and have not cited withdrawn papers.

---

## [Editor Report · Decision Letter 2]

28 Feb 2025

Study on surface deformation induced by shield excavation due to coarse particle content in strongly weathered rock layers

PONE-D-24-45489R2

Dear Dr. Jia,

We’re pleased to inform you that your manuscript has been judged scientifically suitable for publication and will be formally accepted for publication once it meets all outstanding technical requirements.

Kind regards,

Linwei Li

Academic Editor

PLOS ONE

Additional Editor Comments (optional):

None
---

## [Editor Report · Acceptance letter]

PONE-D-24-45489R2

PLOS ONE

Dear Dr. Jia,

I'm pleased to inform you that your manuscript has been deemed suitable for publication in PLOS ONE. Congratulations! Your manuscript is now being handed over to our production team.

Kind regards,

on behalf of

Dr. Linwei Li

Academic Editor

PLOS ONE